# Experiences of Nurses in Nursing Homes during the COVID-19 Pandemic in Germany: A Qualitative Study

**DOI:** 10.3390/geriatrics7050094

**Published:** 2022-09-09

**Authors:** Elisabeth Diehl, David Hiss, Aline Wege, Anna Hirschmüller, Albert Nienhaus, Stephan Letzel

**Affiliations:** 1Institute of Occupational, Social and Environmental Medicine, University Medical Center of the Johannes Gutenberg University Mainz, Obere Zahlbacher Str. 67, 55131 Mainz, Germany; 2Department for Occupational Medicine, Hazardous Substances and Health Science, Institution for Accident Insurance and Prevention in the Health and Welfare Services (BGW), Pappelallee 33/35/37, 22089 Hamburg, Germany

**Keywords:** COVID-19, Corona, SARS-CoV-2, stresses, resources, health care workers

## Abstract

(1) Background: The aim of this qualitative study was to examine pandemic-related changes in nursing work in nursing homes, the resulting work-related stresses and external as well as internal alleviating measures. (2) Methods: We conducted 10 interviews from March to June 2021 with nurses from eight facilities. Data were analysed according to qualitative content analysis. (3) Results: Nurses faced increased workloads due to regulations and guidelines paired with staffing shortages. Work became more difficult due to personal protective equipment (PPE), conflict with residents’ relatives and, in the case of outbreaks, excess death and suffering. Nurse-to-resident care work became more emotionally demanding, with residents more distressed due to the lockdown, while families and social workers were not allowed into the facility. Residents with dementia posed an additional challenge, as they did not remember hygiene and distancing rules. Internal and external measures were not sufficient to alleviate the situation. However, some measures, such as training programmes or existing palliative care concepts, were considered helpful. (4) Conclusions: Facing other possible upcoming pandemics, ways to improve facility administration to prepare for future pandemics are highly needed, such as regular training programmes to prepare for possible lockdown scenarios, PPE use or potential hygiene measures.

## 1. Introduction

In late 2019 and the early stages of 2020, the world was first confronted with the severe acute respiratory syndrome coronavirus 2 (SARS-CoV-2). By March 11, the WHO had declared the Coronavirus Disease 2019 (COVID-19) a pandemic [1]. While the pandemic affects almost everyone’s daily lives due to restrictive hygiene, quarantine and social distancing measures (among others), the worldwide economic slowdown as well as an increased risk of catching a potentially fatal disease [2], health care workers and nurses are directly and disproportionately affected [3]. In an attempt to assess their situation, several studies have been conducted focusing on nursing staff in hospitals and Intensive Care Units (ICUs). A review [4] summarised that nurses working with patients infected with COVID-19 faced long working hours without proper nourishment, depression, anxiety and fear of infection, staffing shortages, lack of communication with patients, and exhaustion as well as a lack of adequate personal protective equipment (PPE).

However, geriatric nurses in nursing homes remain underrepresented in the data, despite being highly impacted by the COVID-19 pandemic [5]. Nursing homes are public or private residential facilities with a focus on long-term care for those who are unable to take care of themselves, which includes mostly the elderly but also, e.g., the chronically ill [6]. The relative sparsity of data is remarkable; as compared to other age groups, the elderly are far more likely to become sick, require hospitalisation, and die when infected with COVID-19 [7]. In Germany, 86% of deaths occurred in the population aged 70 years or older [8]. Thus, nurses in nursing homes are caring for the most vulnerable members of our population. Comment [9] and newspaper articles [10] describe dramatic situations in nursing homes, which are seen as the “largest hotspots for the dissemination and mortality of the COVID-19 pandemic throughout the year of 2020” [11,12].

Internationally, some studies have examined the situation in nursing homes, revealing several common issues. Firstly, the pandemic confronted nursing homes with a novel situation, not only because of the virus itself but also because new and repeatedly changing regulations and guidelines drastically altered nurses working environment [13,14,15]. Furthermore, nurses were afraid of catching COVID-19 themselves or transmitting it to residents or to their own families [13,14,16]. Some studies [13,17,18] reported an initial lack of PPE, which meant that nurses often had to reuse it, increasing the risk of contracting the virus or passing it on. Further challenges arose locally from difficult interactions with residents and their families, especially due to anger at distancing measures on the part of residents and their relatives [15]. Similarly, difficulties arose with residents with cognitive impairments, who neither understood nor subsequently adhered to hygiene and social distancing measures, making enforcing them more challenging [13].

These observations provide an important overview of the situation of nurses in nursing homes during the early stages of the pandemic. However, some researchers were solely informed by head nurses and managers [17,18]. This narrows the scope of problems that can be assessed, as these groups often encounter a different working structure than nurses mainly entrusted with nursing duties. Furthermore, most studies quoted (with the exception of 14) were conducted before major outbreaks had occurred within facilities. Lastly, most studies focused on stresses and issues resulting from the COVID-19 pandemic. While this approach is useful for gaining a broad view of problems, no studies have looked at how the pandemic situation affected day-to-day nursing work specifically. Few studies have focused on intrinsic and institutional coping mechanisms (e.g., good teamwork, training programs) [15,17], so we particularly wanted to know what exactly was being performed to improve the situation of nurses in nursing homes.

The aim of this qualitative study was to explore pandemic-related changes in nursing work and the resulting work-related stress as well as relevant alleviating measures, as reported by geriatric nurses in nursing homes in Germany who were on active nursing duty during this time. There will be a specific focus on stresses and issues resulting from outbreaks as they are so far underrepresented in the data.

## 2. Materials and Methods

### 2.1. Design

This study adopted a qualitative approach with semi-structured interviews. Transcribed interviews were analysed according to qualitative content analysis [19].

### 2.2. Participants

Purposive sampling was used to select participants. Since the interviews were to be conducted during the pandemic, meaning that the nurses had already been exposed to increased stress for over a year, it was assumed during the planning of the study that the nurses’ willingness to be interviewed would not be high. Therefore, 10 interviews were initially targeted. Subsequently, the aim was to determine whether data saturation [20] had been reached or whether further interviews should take place. After eight interviews, no new themes could be gleaned from the interview material. Therefore, it was assumed that data collection had reached saturation point. We continued data collection for all further scheduled interviews to ensure and confirm that no new themes emerged.

First, 2 provinces and 2 cities in Germany (state of Rhineland-Palatinate) were selected. These reported the highest number of COVID-19 cases per 100,000 inhabitants at the reporting date of 21 January 2021 (data retrieved from ”Robert Koch-Institut: COVID 19-Dashboard”, [8]). Then, a list of facilities in these areas was compiled. Out of 58 facilities, 10 were randomly selected. Of those, 5 facilities decided to participate. Another 5 facilities had to be followed up. First, we contacted selected facilities by e-mail. If they did not reply within a week, we phoned them to request their participation in the study. If desired, we sent another e-mail detailing the study’s purpose and its ethics and privacy policy. Usually, we first spoke to the facility manager, who passed on our request to their employees. When a member of the nursing staff who was directly involved in taking care of residents during the COVID-19 pandemic agreed to an interview, the participant received a detailed description of our study and its objective, as well as information on our privacy policy and informed consent by post. If desired, additional information was provided by phone. Participants were asked to sign the informed consent form and mail it back to us using a stamped envelope we included in our correspondence.

From two facilities, two nurses were asked to be interviewed respectively. After one interviewee repeatedly postponed her appointment due to sick leave, and data saturation had been considered reached, we excluded one facility from our examination. Moreover, during one interview, it turned out that the participant worked as a facility manager only and had not been involved in taking care of residents. Therefore, data from this interview were disregarded from further analysis. In the end, 10 interviews with nurses from 8 facilities were considered for further analysis. Recruitment and interviews were conducted from March through June 2021.

### 2.3. Interviews

The interview guidelines were developed based on a literature review and improved through pre-tests with two nursing professionals from nursing homes. Questions were aimed at gathering information on changes in everyday working life due to the pandemic, measures and their evaluation, and mental and physical health problems as well as new or exacerbated stresses and issues. Participants could choose to be interviewed by phone or by video call. Most interviews were conducted by phone (half at nurses’ workplaces and half at their homes, without the presence of third parties). All interviews were recorded and subsequently transcribed. Three qualified members of the research team (ED, DH, AW) conducted the interviews. First, demographic questions were asked, followed by the questions presented in excerpts in Table 1.

### 2.4. Analysis

We performed a qualitative content analysis according to Mayring [19]. A deductive-inductive approach was followed. Deductive main categories were generated through a literature review and formed the basic framework of the analysis. Inductive subcategories were derived from the text analysis (including the possibility of additional inductive main categories) and iteratively refined. Two qualified team members (DH, AW) evaluated the material independently of each other. Subsequently, they compared their results. If they disagreed about categorisation of a text element, they first attempted to form a consensus. If no consensus could be reached, the material in question was presented to the project team as a whole, where it was discussed until a decision was reached. Interview quotes were translated into English with the support of an English native speaker for publication purposes [21], and the COREQ checklist (Consolidated criteria for reporting qualitative research) [22] was used in order to assure the reporting quality of our study.

## 3. Results

### 3.1. Participants

The participants’ demographics are shown in Table 2. Interviewees were aged 29–59 (*M* = 47.4, *SD* = 8.9). Our sample included nine trained nurses and one care assistant. The duration of the interviews ranged between 30 and 73 min (*M* = 47.7, *SD* = 15.6).

### 3.2. Changes in Day-To-Day Nursing Work in Response to the Pandemic Situation

Issues identified by the nurses concerning their work life that resulted from the pandemic are listed in Table 3 and described in the following paragraphs.

#### 3.2.1. Isolation and Lockdown

New guidelines and regulations concerning COVID-19 had a tremendous impact on nursing work. Distancing rules dictated a minimum distance between two individuals in order to minimise the spread of COVID-19. In nursing homes, the only way to comply with these rules was to isolate residents. At the beginning of the pandemic, all facilities were on lockdown. In time, local governments loosened these rules, but a lockdown was still often necessary, e.g., whenever an outbreak occurred. The effect of these measures on residents strongly differed from person to person. However, it nonetheless changed the nursing work. 

“*So, we had some residents who were notably calmer than usual. […] It was really very notable—in my opinion—because they did not have to, well, see their relatives every day and for some of them that was a relief, really. Others, on the other hand, really missed them. They were sad often, cried a lot. So, we had to be there socially*.” *[I. 7, residential sector manager, COVID-19 outbreak in facility]*

Specifically, nursing work changed in the sense that nurses had to be more flexible in their interaction with residents. As families and social workers were not allowed inside the facility, nurses often had to fill in these blanks, which was even harder because many daily activities with residents were restricted due to distancing regulations.

“*During the pandemic, especially while relatives could not visit or during the outbreak nobody could visit either. During that time, we were not only nurses, we were everything: relatives, social workers, everything*.” *[I. 7, residential sector manager, COVID-19 outbreak in facility]*

In addition, isolation meant that, in many cases, residents did not eat in the dining hall but in their own rooms. Nurses also had to bring food to each room, which added to time constraints; this, too, had an impact on nursing work since psychological mechanisms that had motivated residents to eat when in company ceased to exist.

“*These things [less group activities and eating alone] changed every-day life a lot. Residents who ate normally when in company—because of this mirror-reflection, you know: he sees, ‘oh, everyone is eating, so I eat too’, that was not there anymore. And that was difficult of course, because no number of nurses could have compensated what we lost in this community*.” *[I. 7, residential sector manager, COVID-19 outbreak in facility]*

This intensified care work was mostly solved by nurses working overtime.

“*Well, it was kind of stressful. Yes, overtimes as well. By December I had more than 90 h of overtime*.” *[I. 2, residential sector manager, COVID-19 outbreak in facility]*

Throughout all interviews, a recurring theme was that residents showed more signs of depression, weariness of life, as well as worsening dementia during the pandemic, especially during periods of lockdown. This had consequences for nurses’ job descriptions as well:

“*They had no access to social support or social support groups. And so, we had to compensate for that as well and not only take on nursing duties but also take care of emotional support. That was a burden some time for us as well, you know*?” *[I. 4, residential sector manager, no COVID-19 outbreak in facility]*

#### 3.2.2. Frequently Changing Guidelines and Regulations

At the beginning of the pandemic, guidelines and regulations were constantly being updated. For months, facilities faced ever-changing rules. Nurses we interviewed indicated that working under such circumstances was difficult.

“*You had to adapt to new rules every day. What can I do, what can’t I do, what do we have to do, what rules do we have to stick to now, you know? That was pretty difficult, really difficult. I have worked as a nurse for 30 years and have never experienced anything like this*.” *[I. 1, residential sector manager, COVID-19 outbreak in facility]*

#### 3.2.3. Outbreaks, Sickness, Excess Deaths and Suffering

Nurses described outbreaks of COVID-19 as uncontrollable. Infection rates among residents and staff grew day by day, while not much could be performed, meaning nurses felt powerless to counter this spread.

“*At some point, we were just powerless. Yes, that is the right word. And every morning we hoped, hopefully today it won’t get worse or more are infected*.” *[I. 10, nurse, COVID-19 outbreak in facility]*

Frequent death among residents is an inherent aspect of the job of a geriatric nurse. However, during the pandemic and especially when an outbreak occurred, there were far more deaths than usual. The many residents suffering from a deadly respiratory disease at the same time made taking care of residents with COVID-19 for nurses physically difficult and emotionally frustrating. 

“*Within four weeks, we had as many deaths as we would normally have within a year. Some of these residents had lived here for a very, very long time and of course this didn’t leave staff unscathed, it really upset many of them*”.—*[I. 7, residential sector manager, COVID-19 outbreak in facility]*

“*And no matter what you do, how much effort you put in, the person still dies. And they die—how can I explain… Like dry leaves falling from a tree, one after the other. Ambulances and hearses were parked at the entrance every day. It was like they came to work with us. […] Yes, we accompany these people until the end. That is just what we do here. But suddenly it’s like one after the other*…” *[I. 8, nurse, COVID-19 outbreak in facility]*

On top of that, nurses were confronted with severe suffering among residents infected with COVID-19.

“*Of course, I don’t want something like Corona to happen again, that people have to suffer. You know, because, their gaze, mouth open… And in their mouth, everything is black […] they want to drink but can’t swallow because they can’t breathe and their throat is dried up because their mouth is open constantly […] their eyes stare at the ceiling, it’s like in a horror movie*.” *[I. 8, nurse, COVID-19 outbreak in facility]*

To cope with this intensified care work, a nurse reported how they attempted to streamline workflows when working with the sick. 

“*So, we tried to streamline workflows a bit. Like, for example, our sick, they will stay in bed anyway. Therefore, it is not so bad, if we do not wash them at 8 a.m. during breakfast, so they can have their breakfast in peace and then we can get them ready and take care of them the rest of the day. […] So, we said, basic care that we usually do by eight, nine a.m., let us do this in a 24-h-rhythm, regardless of whether it is morning or afternoon. So, we adapted the workflow a bit*.” *[I. 11, head nurse, COVID-19 outbreak in facility]*

#### 3.2.4. Own infection with COVID-19 and fear of infection in self and others

Many nurses were infected with COVID-19—sometimes with very severe symptoms and some of them with severe long-term effects.

“*Yes, of course, many of us were infected. In some it was barely notable, in some more than in others. But we had some cases with very, very bad symptoms, who to this day struggle with symptoms and they were infected in the beginning of January [note: four months before the interview]*.” *[I. 7, residential sector manager, COVID-19 outbreak in facility]*

Even if no infection occurred, nurses in several interviews reported severe psychological pressure due to the fear of being infected. 

“*Yes, that is right and when colleagues who suffer from pre-existing conditions just heard someone coughing, getting a cold, it made their hair stand on end. So, yeah, I think it will take some time until we all processed this, really*.” *[I. 6, nurse, COVID-19 outbreak in facility]*

Of course, being infected oneself was not the only problem. Due to the high transmissibility of COVID-19, most nurses reported being afraid of being the cause of infection in others, such as their families or residents. Nurses were especially concerned about being the cause of an outbreak in their facility.

“*We tried to stick to all the rules but then, you see, some have symptoms. Some are tested positive and they have zero symptoms. So, I always said, we just don’t know what happens, when in the end, there is an outbreak… and then we get tested and it turns out, it’s you who brought it in. […] You didn’t want to be the one who brought it in*.” *[I. 5, nurse, COVID-19 outbreak in facility]*

#### 3.2.5. Staff Shortages

Infection created another problem: staff shortages.

“*The staffing situation is like that, that often you do not have as many nurses as you would need. Like, on our floor we have 35 residents and if we are staffed well, we are four nurses in the morning, which is not enough […]. During the pandemic, this was of course worse, like in the morning we were three instead of four nurses. That did not make it any easier*.” *[I. 7, residential sector manager, COVID-19 outbreak in facility]*

“*Then my co-workers got sick. I think we had 15 staff members who got sick with Corona. In my ward, they were five and we had to fill in these blanks somehow, you know*.” *[I. 1, residential sector manager, COVID-19 outbreak in facility]*

#### 3.2.6. Time Constraints

With the emergence of the pandemic, time constraints increased for several reasons:

“*Working procedures changed, yes, and this meant that all residents are in their rooms and not in the dining hall. This meant we had to go to their rooms more often and many tasks came on top of that, like regular control of vital signs that we had to do on top of everything else*.” *[I. 2, residential sector manager, COVID-19 outbreak in facility]*

There were other time-consuming factors, such as frequent testing of residents, staff and visitors.

“*Testing is still… Once a week, that is very intense. Two times a week—I wonder how we were ever able to manage that. Well of course. By people staying longer, saying, ’I’ll do some tests after my shift‘. It would not have been possible otherwise*.” *[I. 3, head nurse, COVID-19 outbreak in facility]*

The need to take PPE on and off whenever visiting another resident, as well as the constant need to disinfect, added to time constraints.

“*Well, it was not so great, just think of how we had to wrap ourselves up each day in full gear with masks, gowns and who knows what and you sweat under those things and then: from room to room, always on and off and once you’re out disinfecting your hands* …” *[I. 10, nurse, COVID-19 outbreak in facility]*

#### 3.2.7. Problems due to PPE

The requirement to wear PPE was not only a time constraint. Many nurses pointed out that wearing PPE made working more difficult. One nurse indicated that constant mask-wearing across hour-long shifts affected her respiratory health and that of her colleagues.

“*It’s just like, obviously everyone is used to it by now and of course we do it [wearing masks], but you can tell… After the shift when you leave the nursing home, take of the mask. For me, by now my nose is totally defective, others get a dry throat. You need about an hour and a half until you are back to a level, where you can say, ’OK, I can breathe again’*”. *[I. 5, nurse, no COVID-19 outbreak in facility]*

Another more frequent complaint about wearing PPE was heat stress. This was true not only for hot days:

“*Most resident’s rooms are very well heated and if on top of that they shower, that is… Sometimes you just think, ’Ok, I just want to rip this mask off my face so I can gasp for air, because [breathes deeply] it’s lacking’*”. *[I. 5, nurse, no COVID-19 outbreak in facility]*

However, none of the nurses we interviewed questioned the necessity of wearing PPE as such. What they did describe as problematic was a PPE shortage in the early stages of the pandemic, especially when it came to masks.

#### 3.2.8. Residents with Dementia

Residents with dementia posed specific problems during the pandemic that require further explanation:

“*Well, I mean… with the demented it was one of those things… All residents had to be in their rooms, right? Now try to explain that to someone with dementia, who does not even understand you and who in turn leaves his room and walks back and forth or into other rooms. Maybe even tested positive*”. *[I. 10, nurse, COVID-19 outbreak in facility]*

Asked how they dealt with the problem, the nurse replied:

“*Well. How? You cannot lock them up. We tried to interfere occasionally when we saw it happening. Otherwise, this is just where you reach your limits. How could you do it? […] And what is even worse: he will not keep the mask on. […] The risk was there constantly*.” *[I. 10, nurse, COVID-19 outbreak in facility]*

Other facilities solved this problem through one-to-one care, which of course, was resource-intensive:

“*The only way to manage this is, by scraping together everything that moves. […] We had to arrange it so that a nurse was present at least between seven in the morning […] until they went to sleep*.” *[I. 3, head nurse, no COVID-19 outbreak in facility]*

#### 3.2.9. Communication

Communication between nurses and residents changed for multiple reasons. One problem resulted from masks, as residents with hearing problems had difficulties understanding nurses due to muffled speech and the missing possibility of paying attention to lips.

Often, the only way to solve this problem was to remove the mask for an instant. 

“*They do not see the mouth moving or cannot hear the sound of our voice. Many recognize us just from speaking. They know exactly this is nurse [name removed], you know, but with the masks, all this changed. Yeah, so sometimes you really have to take the mask off to show who you are and then put it back on*.” *[I. 1, residential sector manager, COVID-19 outbreak in facility]*

Another factor that affected nurses’ communication with residents but also their relatives was the pandemic as a whole and explaining this problem, sometimes to people with cognitive impairment.

“*You had to explain to residents what was happening around the world. And that required tact, especially when there were lockdowns everywhere, when the facilities were closed too. To explain this was incredibly difficult when it came to relatives but also when it came to residents. So, this was notable, that you were confronted with wholly different aspects of geriatric nursing*.” *[I. 11, head nurse, COVID-19 outbreak in facility]*

#### 3.2.10. Conflicts with Relatives

Another issue that in many facilities became more difficult was contact with residents’ relatives. Initially, they were not allowed to see their relatives residing in nursing homes. In the later stages, they were allowed in if the facility was not on lockdown. However, regulations required them to wear masks and to be tested. Many nurses reported increased conflict with relatives, some of whom did not understand the need for regulations and guidelines and sometimes did not adhere to them. 

“*They came without masks, even though there were signs outside. They did not register, even though they had to, so in the end, you could not let them out of your sight and doing this alongside your work caused some additional stress*.” *[I. 5, nurse, no COVID-19 outbreak in facility]*

### 3.3. Internal and External Measures to Alleviate Stresses and Issues Resulting from the Pandemic

One research question addressed attempts to alleviate the situation of nurses in nursing homes amid the COVID-19 pandemic, along with an evaluation of whether such attempts were helpful. Generally, measures were taken on two levels: internally, i.e., within the facility or externally, e.g., when help arrived from outside the facility. The measures are listed in Table 4.

#### 3.3.1. Internal Measures—Training Programmes

It appears that not all facilities offered training programmes for handling COVID-19 or that they were not always reported. However, where training programmes of any kind were implemented, they were usually described as helpful. 

“*We still practice once a month. I say: ‘imagine Mrs. M. is positive, what do we do?’ And we practice until Corona is over and probably until I retire*.” *[I. 3, head nurse, no COVID-19 outbreak in facility]*

One type of programme that was described as especially helpful during the COVID-19 pandemic was palliative support.

“*Let me put it this way: In our nursing home, we are very well equipped when it comes to training programs. We have a well-working palliative concept that I can only recommend to all nursing homes—trained palliative care professionals, who do not only have an eye on residents but also on the staff. With all of these deaths, this was particularly helpful*.” *[I. 7, residential sector manager, COVID-19 outbreak in facility]*

#### 3.3.2. Infrastructural Changes

In some cases, infrastructure was adapted to the pandemic situation. Where infrastructural changes were possible and performed, they were considered helpful, i.e., because they allowed infected residents to be separated from the rest of the ward.

“*In our house, we could structurally manage to set up a quarantine station. […] we were able to set up two rooms with 3 beds each and we set them up in such a way that it really works independently […] of the rest of the house*.” *[I. 11, facility manager, COVID-19 outbreak in facility]*

#### 3.3.3. Internal Support

In many facilities, personnel from other areas within the facility support nursing staff.

“*So, we asked staff from housekeeping, like, ‘can you do us a favour? Like make some beds or something?’ With activates like these, housekeeping supported us a bit. Social care staff helped us in distributing food and drink, because normally we say, they should not be involved when it comes to food; it just is not their job. But of course, everyone helped each other. So, the whole facility worked together*.” *[I. 11, head nurse, COVID-19 outbreak in facility]*

#### 3.3.4. Motivational Measures

In some cases, facility management attempted to support nurses by means of motivation and recognition. This ranged from e-mails from management thanking nurses for their hard work to gift baskets and small financial incentives. 

“*Yes, management constantly or often tried to motivate us, thank us or told us to hang in there*.” *[I. 10, nurse, COVID-19 outbreak in facility]*

#### 3.3.5. Team

Almost without exception, interviewees indicated that team cohesion had improved throughout the pandemic. Nurses reported an atmosphere of mutual support as well as comradery among nursing staff in the face of very difficult circumstances.

“*Yes, sometimes we could laugh, yes. Even though we often cried together, there were times or phases when we really could just laugh and that is… that is worth a lot if you can laugh despite everything that happened in here. Yes, that is worth a lot*.” *[I. 6, nurse, COVID-19 outbreak in facility]*

“*Us nurses, who were in action constantly… We were a little bit… I mean, there was team spirit, there was a sense of cohesion. That was just something special, yes*.” *[I. 10, nurse, COVID-19 outbreak in facility]*

#### 3.3.6. External Measures—Staffing Support

During one phase, German soldiers were recruited to work in nursing homes. However, reviews were mixed. Generally, this measure was perceived as helpful. 

“*Yes so, during a period of three or four weeks, we had one person from the military per ward. They supported us with things like refilling drinks in the rooms, so they had no nursing duties in that sense but rather took care of residents’ well-being so that we had more time for nursing work. So, they refilled water, hung up towels, made the beds, distributed food. Those were the things they did. [Question: Did that help?] Yes absolutely*.” *[I. 7, residential sector manager, COVID-19 outbreak in facility]*

However, in some cases soldiers were said to have arrived at the wrong time, not during an outbreak when help was needed the most.

In other cases, no external support arrived, even though it was requested. 

“*And then the support with testing… […] ‘Yeah, OK, we’ll send support’. But nobody came, no matter where we asked. It really is a joke*.” *[I. 3, head nurse, no COVID-19 outbreak in facility]*

Two nurses from facilities where no help had arrived were sceptical about the concept as such, as seeing strangers would have been additional stress for residents.

*[Question: Would you have liked external support?] “Honestly, no. Because residents know us. If on top of everything, strangers had come, that would have been… That would not have been nice for them*.” *[I. 4, residential sector manager, no COVID-19 outbreak in facility]*

Apart from the support of the military, some facilities managed to find external support from other sources.
“*And then, we placed an ad in our local journal and explained our situation and looked for volunteers to support us. And I think one or two even came regularly. Every day for about two hours. And they just supported us in the residential area, did some room care, refilled drinks and such. That was really nice, yes*.” *[I. 11, head nurse, COVID-19 outbreak in facility]*
In another case, support arrived from retired nurses who had formerly worked in the facility.

#### 3.3.7. Financial Recognition

Aside from staffing support, politicians and the public tried to keep up nurses’ morale. Primarily, the government opted for a “Corona-Bonus”. While opinions on the one-time payment of up to EUR 1500, provided by the German government, varied across the board, nurses stressed that this was not a long-term solution.

“*Yes, we received it in December. Well, the tenor in the facility is that it’s nice to have but it won’t change anything about working-conditions and next month I will once again have my meagre salary*”. *[I. 3, head nurse, no COVID-19 outbreak in facility]*

One nurse indicated that the bonus was just enough to pay the childminder she had to employ because of additional work that resulted from the COVID-19 pandemic and overall did not think it was appropriate compensation.

“*Yes, good timing, that way I could pay my childminder. Because she took care of my daughter seven days on end for 24 h, because I had a night shift that I couldn’t swap and there was no one else. So yeah, in relation to what we did, I thought it was a joke. Sorry for saying it like this*.” *[I. 6, nurse, COVID-19 outbreak in facility]*

#### 3.3.8. Motivational Measures

In the early stages of the pandemic, the German public coordinated a campaign where people went out onto their balconies to applaud health care workers for their commitment. However, the nurses we interviewed had mixed feelings about this. One nurse indicated that she was surprised that they suddenly received recognition, while for a long time, she had felt that nobody noticed her professional group’s struggles.

“*What I do not understand is: all these years nobody really cared about us, neither in hospitals nor in geriatric care and now there is a pandemic and—I don’t know —all of a sudden, everyone claps and compliments you and you are recognized. Well, and actually it should be normal that those who care for the old and sick are respected, because it would be adequate*.” *[I. 6, nurse, COVID-19 outbreak in facility]*

Another nurse pointed out that this form of recognition in the sense of a bonus or applause is meaningless to her if the strenuous nature of the work means she has to retire early, which in turn reduces her pension.

“*The problem is, if I retire with 63 and manage until then, I will be broke and then I will have reduced pension. After more than 41 years as a nurse, so… but nobody cares. […] €1000 are of no use to me, if someone puts a candle in their window or claps, it is of no use to me. They can keep that to themselves*.” *[I. 3, head nurse, no COVID-19 outbreak in facility]*

Generally, there seemed to be a sense that the recognition disappeared as quickly as it had appeared.

“*And then they slip you €1500 and you get a pat on the back, like ’you are the best, thank you‘. And now you hear nothing anymore, about us or about the nurses and doctors who all had to run themselves to death*.” *[I. 8, nurse, COVID-19 outbreak in facility]*

## 4. Discussion

The reported experiences of nurses in nursing homes during the COVID-19 pandemic in Germany support findings from previous studies around the world. Nurses reported facing a novel situation [15] with very different working conditions due to the new virus as well as new guidelines and regulations that, in some cases, changed on a weekly or even daily basis [14], at least during the early stages of the pandemic. Furthermore, this meant an increased workload [17], e.g., due to the requirement to test every resident, staff member and visitor [23], while at the same time, some nursing homes faced staffing shortages due to infections with COVID-19 as well as quarantine regulations [24]. New guidelines and regulations also made work more difficult [25], for instance, due to mandates for the wearing of masks, which cause respiratory difficulties or heat stress. This aspect has been poorly discussed in the literature, as during the earlier stages of the pandemic, the availability of PPE was the dominant problem [13]. In sum, their workload was increased, and the work itself became more challenging, while there were fewer staff to do it. On top of that, nurses reported increased psychological pressure due to fear of being infected themselves or being the cause of an outbreak [14].

Diverse studies address the difficulties of caring for patients with dementia in nursing homes during the COVID-19 pandemic [14,26]. Like our study, these show that residents with dementia presented a special challenge, as they did not remember mask mandates or social distancing guidelines. If they were infected with COVID-19, this was especially problematic, as they threatened to infect other residents. Nonetheless, so far, how facilities dealt with this has not been described: our interviews showed that nursing homes with sufficient resources were able to provide residents with dementia with one-to-one care when they were infected. Where fewer resources were available, sometimes there was no choice but to hope to catch them in time when they wandered the hallways.

Nurses’ job descriptions changed profoundly during the pandemic. This was especially true during times of lockdown during the first wave of COVID-19, where governments around the world implemented visitor bans in nursing homes, meaning that relatives and social workers, as well as diverse service providers, could not enter the facilities [27]. During these periods, nurses reported trying to replace them as best they could. Generally, they had to provide more emotional support than usual, as residents showed symptoms of depression and anxiety, and some even lost the will to live, a fact that was also reported in studies around the world [14,27,28]. Many of the protective measures, such as distancing rules and mask mandates, were taken to protect the residents. However, these measures’ long-term psychological and physical consequences for the residents have hardly been studied [29]. Yet it is precisely these long-term psychological and physical consequences that should be studied and discussed further in the future to ensure better preparedness for future pandemics. For example, in the current recommendations of the Robert Koch Institute for elderly care and nursing facilities [30], if several SARS-CoV-2-positive cases or suspected cases occur in a facility, joint isolation of several residents with permanently assigned staff is recommended. However, the spatial and personnel possibilities must be available for this, and the latter especially seems to be difficult to implement against the background of the nursing shortage, which is not only in Germany [31].

However, as most previous studies were conducted during earlier stages of the pandemic, little is known about stresses and issues caused by outbreaks of COVID-19 in facilities. Outbreaks had several severe consequences, as a large proportion of deaths from COVID-19 are related to outbreaks in nursing homes [32]. Many studies showed that geriatric nurses faced an excess of deaths in residents [11]. Less frequently discussed, however, were the consequences of these numerous deaths for the nursing staff. From the interviews, it emerges that, in principle, it must not be forgotten that nursing staff from nursing homes often have a long relationship with the residents, sometimes caring for them for years, and often know their relatives and friends. In the event of an outbreak, they had to cope with not only multiple resident deaths at the same time but in many cases, their patients’ profound suffering, which they were unable to alleviate or cure. In the future, it should be investigated in more detail to what extent experiences such as these have an impact on the mental health of nurses in geriatric homes and how nurses can be helped in the best possible way in the future to deal with such situations and to process these experiences. International studies show that palliative care nurses have more social, personal and organisational resources than nurses in other fields [33,34,35,36]. These resources contribute to palliative care workers not only being better able to cope with the stresses of death and dying but also being healthier and more satisfied with their profession. Consequently, it is not surprising that this study shows that palliative support, in particular, was seen as especially helpful in dealing with both the numerous deaths and the agonising circumstances of death. Against the background that more and more nursing staff are thinking about leaving their jobs [37], in the future, it should be evaluated which aspects of palliative care—and this not only for exceptional situations such as the SARS-CoV-2 pandemic but in general—are considered particularly useful for nursing homes in order to be able to cope better with the workloads.

The results of our study suggest that facility-internal alleviating measures were often born out of necessity in the moment and were in no way able to mitigate increased stresses. External measures did help in some cases, but overall, support was too little to alleviate the situation sufficiently. While military support was considered helpful where it arrived, it was often felt that it did not arrive in time. In some cases, it was promised but not delivered. As for the “Corona-Bonus”, especially where an outbreak occurred, nurses felt this one-time payment did little to compensate for what they had lost in terms of quality of life and personal health or their families’ health during the pandemic. Most nurses stressed that this one-time payment did not make up for their low salaries and, therefore, their financially dire situation in the long run. Motivational support, such as applauding nurses’ efforts, was not perceived as helpful. There seemed to be a sense that respect for (geriatric) nursing was lacking before the pandemic, briefly peaked during the early stages of the crisis, and ebbed away soon after. Generally, nurses wished for a long-term solution, stressing that respect is best expressed through higher salaries and improved staffing. Additionally, this study supports the findings of other studies, which showed that there were measures that were unequivocally considered helpful: training programmes [15,25,38]. First, training programmes prepared nurses to deal with infections and outbreaks and, in that way, provided a sense of security. In addition, palliative support, as already discussed above, was described as invaluable for coping with excess deaths and the difficult situation in general. It should be stressed that this programme was available in a facility where it had already been available pre-pandemic. The authors suggest that it is difficult to install such infrastructure during a pandemic. In addition, there are both international and national recommendations and guidelines for nursing homes with numerous strategies designed to prevent the occurrence and spread of COVID-19 disease within a facility [30], and various studies have now also been conducted about guidelines and recommendations for the support of health care workers during the COVID-19 pandemic. A scoping review from the year 2021 clustered the guidelines and recommendations into four main categories: “social/structural support,” “work environment,” “communication/information”, and “mental health support”. The authors of this review criticised the fact that not only is the empirical evidence on the effectiveness of these recommendations missing, but also that most recommendations were developed without involving different members of the target group, such as nurses or other stakeholders [38]. In future, more efforts must be made to develop strategies that are suitable for everyday use and are evidence-based, to provide nursing homes and caregivers with better, faster and more targeted support, especially in crisis situations.

### Limitations

Results in the present study, although giving an important direction for further inquiry, are not generalisable in themselves due to their qualitative nature. There may be a response bias, as some facilities refused to participate in the study. As we initially communicated with facility management, there is a possibility that facilities that had faced greater difficulties were also more likely to refuse participation, so it is possible that the present results, while painting a dramatic picture, may not include the worst cases. On the other hand, the process of recruitment entailed contacting nursing homes in provinces with a high incidence of COVID-19 cases. We performed this to increase the likelihood of picking facilities that had faced an outbreak. However, facilities with outbreaks may be overrepresented in the present study. Lastly, our recruiting was restricted to the area of Rhineland-Palatinate, and findings are not necessarily valid for other parts of Germany and much less the world.

## 5. Conclusions

In sum, the results support findings from previous studies and add further information about the situation of nurses in nursing homes during the pandemic.

Taken together, our account of stresses and issues paints a troubling picture where pre-existing difficulties such as time constraints and staffing shortages increased and new stresses such as extended use of PPE were added.

Our look at nursing work, as well as alleviating measures, suggests that there is some room for improvement when it comes to handling a pandemic for facilities and also for politicians. Interviews show that training programmes helped nurses to feel more prepared for the challenges of a pandemic. With the current pandemic but also the possibility of future pandemics in mind, we suggest preparing for future emergency situations of a similar nature, as is customary for organisations to prepare for the risk of fire. Such preparation may prove vital, as due to climate change, the risk of zoonotic spillovers and, therefore, future pandemics has grown [39]. Facilities could achieve better preparation through regular training programmes to prepare for possible lockdown scenarios, PPE use or potential hygiene measures, even outside of a pandemic. This way, in the future, nurses could enter a difficult situation with a sense of preparedness. COVID-19 also confronted nurses with a sharp increase in deaths. Interviews suggest that a palliative concept was helpful regardless of, but especially during, the pandemic. While far from inexpensive, implementing a palliative concept in more facilities could be an important factor in improving working conditions as well as preparing for a possible future pandemic. A vital part of palliative programmes concerns, for example, self-care training for nurses. These training sessions focus on supporting nurses to increase coping skills and resiliency by helping them reconnect with the reasons why they chose their profession in the first place. Programmes such as these could be freely available online as they already are for the English-speaking community [40].

Lastly, most nurses stressed that the two most pressing problems were staffing shortages and low salaries before and even more so during the pandemic. Politicians need to address these issues quickly and decisively. The pandemic is far from over, and it may take time until COVID-19 can be considered endemic [41]. This crisis laid bare the vulnerability of health systems around the globe [42]. With more nurses deciding to quit their jobs and young talent lacking due to people often opting for professions with better working conditions [31], the situation of nurses in nursing homes and also the health system, in general, is likely to become even worse. In the end, failure to provide nurses in nursing homes and other areas with adequate working conditions may negatively affect not only nurses themselves but anyone who may require their services now and in the future.

## Figures and Tables

**Table 1 geriatrics-07-00094-t001:** Interview guideline (in excerpts) for the conducted interviews from March to June 2021.

Research Question	Question in Interview (in Excerpts)
How did care work change during the COVID-19 pandemic?	Please describe how the COVID-19 pandemic has affected your everyday working life. Please describe how your work organisation has changed. Please describe if new stresses have emerged and if so, what were they?
What measures were taken to alleviate the situation?	What measures were taken, in order to cope with the issues and stresses resulting from the COVID-19-pandemic? ○ How helpful were these measures? ○ What measures do you think would have been useful?

**Table 2 geriatrics-07-00094-t002:** Demographics of participants and participating facilities.

Aspect		Frequency
Profession	Trained nurse	9
	Geriatric care assistant	1
Role	Nursing support worker *	1
	Nurse	3
	Residential sector manager	4
	Head Nurse	2
Gender	Female	9
	Male **	1
Age	20–29	1
	30–39	1
	40–49	3
	50–69	5
No. of residents in facility	50–100	4
	101–150	4
COVID-19 outbreak in facility?	Yes	6
	No	2

* To protect participants’ identities, this participant will be referred to as nurse; ** To protect participants’ identities, all nurses will be referred to as female.

**Table 3 geriatrics-07-00094-t003:** Changes in day-to-day nursing work in response to the pandemic situation.

Issues
Isolation and Lockdown Frequently changing guidelines and regulations Outbreaks, sickness, excess deaths and suffering Own infection with COVID-19 and fear of infection in self and others Staff shortages Time constraints Problems due to PPE Residents with dementia Communication Conflicts with relatives

**Table 4 geriatrics-07-00094-t004:** Measures to alleviate stresses and issues resulting from the pandemic.

Source	Measures
Internal	Training programmes
	Infrastructural changes
	Internal support
	Motivational measures
	Team
External	Staffing support
	Financial recognition
	Motivational measures

## Data Availability

Not applicable.

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
