# Peer review of "Experiences of Nurses in Nursing Homes during the COVID-19 Pandemic in Germany: A Qualitative Study"

_geriatrics, 2022, doi:10.3390/geriatrics7050094_

Round 1
Reviewer 1 Report
1. The manuscript represents a qualitative analysis of nursing reported outcomes in long-term care related to the COVID public health emergency.
2. In the abstract indicate the date that the work was done (June 2021).
3. In the abstract conclusion please provide information about the main messages–perhaps the content of lines 679–680.
4. In table to please indicate the date of the study was performed in (June 2021).
5. Consider placing the list of issues identified by the nurses in a new table (3.2 #1-10).
6. Consider placing the recommendations in a new table (3.3 # 1-8)
Author Response
- The manuscript represents a qualitative analysis of nursing reported outcomes in long-term care related to the COVID public health emergency.
Response: We think this study makes a valuable contribution to the field.
- In the abstract indicate the date that the work was done (June 2021).
Response: As suggested by the reviewer, we have added the date the research was done in the abstract (line 13).
- In the abstract conclusion please provide information about the main messages–perhaps the content of lines 679–680.
Response: We have added the suggested information about the main messages of the study to the abstract (lines 23-26).
- In table to please indicate the date of the study was performed in (June 2021).
Response: Thank you for pointing this out. We have added the date the data was collected in Table 1, Interview guidelines (in excerpts) for the conducted interviews from March to June 2021 (line 130).
- Consider placing the list of issues identified by the nurses in a new table (3.2 #1-10).
Response: We think this is an excellent suggestion. We have put the changes in day-to-day nursing work in response to the pandemic situation into a new table. Table 3 now shows all changes that happened during the pandemic, followed by detailed explanations of the changes in Chapter 3.2 (line 155).
- Consider placing the recommendations in a new table (3.3 # 1-8)
Response: Thank you for this suggestion, we also think a new table would help get a better overview for the reader. We have put the internal and external measures into a new table 4 (line 420), which now shows all the internal and external measures to alleviate stresses and issues resulting from the pandemic. Detailed explanations follow table 4.
Reviewer 2 Report
Thank you for this important and novel work describing the experience of nurses working in nursing homes during the early part of the COVID-19 pandemic.
The study is well conducted and the reporting thorough, although the sample size (and sampling frame) are small. The authors provide a rationale and explanation for this, and the findings are sufficiently valuable that this seems acceptable for a qualitative study with the depth apparent in the manuscript. The limitations and uncertainties are appropriately described.
I have no major concerns about the study and feel it makes an important contribution to the literature. It provides a clear depiction, supported by data of, that resonates with international experience in the aged care sector during this time.
I have some minor concerns regarding the length and use of language in the manuscript, and feel it could be edited for both brevity and clarity. There are more quotes than necessary, which becomes somewhat disruptive to the flow of the narrative. On p10, line 429 a quote seems to be duplicated in a way that is not helpful or suitably illustrative. The discussion is also lengthy and at times feels repetitive of the presentation of results.
Author Response
1. Thank you for this important and novel work describing the experience of nurses working in nursing homes during the early part of the COVID-19 pandemic.
The study is well conducted and the reporting thorough, although the sample size (and sampling frame) are small. The authors provide a rationale and explanation for this, and the findings are sufficiently valuable that this seems acceptable for a qualitative study with the depth apparent in the manuscript. The limitations and uncertainties are appropriately described.
I have no major concerns about the study and feel it makes an important contribution to the literature. It provides a clear depiction, supported by data of, that resonates with international experience in the aged care sector during this time.
Response: Thank you for your detailed feedback on this study. We are pleased that the data collected reflects the experiences of the nurses and we are confident that the insights from this survey will help understand and improve the situation of caregivers.
2. I have some minor concerns regarding the length and use of language in the manuscript, and feel it could be edited for both brevity and clarity. There are more quotes than necessary, which becomes somewhat disruptive to the flow of the narrative. On p10, line 429 a quote seems to be duplicated in a way that is not helpful or suitably illustrative. The discussion is also lengthy and at times feels repetitive of the presentation of results.
Response: Thank you for your feedback and suggestions. We strive for brevity and clarity; therefore, we tried to shorten the manuscript as much as possible, without losing valuable content. We removed some quotes in the results chapter 3.2 and 3.3 (lines 152-562) and revised the discussion (lines 563-671) a bit for a better flow of the narrative and less duplications.